



Earth System
Dynamics

# Human influence on European winter wind storms such as those of January 2018

**Robert Vautard[1], Geert Jan van Oldenborgh[2], Friederike E. L. Otto[3], Pascal Yiou[1], Hylke de Vries[2], Erik van Meijgaard[2], Andrew Stepek[2], Jean-Michel Soubeyroux[4], Sjoukje Philip[2], Sarah F. Kew[2], Cecilia Costella[5], Roop Singh[5], and Claudia Tebaldi[6]**

[1]Laboratoire des Sciences du Climat et de l'Environnement, UMR 8212 CEA/CNRS/UVSQ, IPSL and University Paris-Saclay, Gif-sur-Yvette, France
[2]Royal Netherlands Meteorological Institute (KNMI), De Bilt, the Netherlands
[3]Environmental Change Institute, University of Oxford, Oxford, UK
[4]Météo-France, Direction des Services Climatiques, Toulouse, France
[5]Red Cross Red Crescent Climate Centre, The Hague, the Netherlands
[6]National Center for Atmospheric Research, Boulder, Colorado, USA

**Correspondence:** Robert Vautard (robert.vautard@lsce.ipsl.fr)

**Abstract.** Several major storms pounded western Europe in January 2018, generating large damages and casualties. The two most impactful ones, Eleanor and Friederike, are analysed here in the context of climate change. Near surface wind speed station observations exhibit a decreasing trend in the frequency of strong winds associated with such storms. High-resolution regional climate models, on the other hand, show no trend up to now and a small increase in storminess in future due to climate change. This shows that factors other than climate change, which are not in the climate models, caused the observed decline in storminess over land. A large part is probably due to increases in surface roughness, as shown for a small set of stations covering the Netherlands and in previous studies. This observed trend could therefore be independent from climate evolution. We concluded that human-induced climate change has had so far no significant influence on storms like the two mentioned. However, all simulations indicate that global warming could lead to a marginal increase (0 %–20 %) in the probability of extreme hourly winds until the middle of the century, consistent with previous modelling studies. This excludes other factors, such as surface roughness, aerosols, and decadal variability, which have up to now caused a much larger negative trend. Until these factors are correctly simulated by climate models, we cannot give credible projections of future storminess over land in Europe.

## 1 Introduction

The influence of climate change on extratropical storms has been the subject of a number of studies so far (Ulbrich et al., 2009). It has been demonstrated that with the expansion of the Hadley cell the jet streams and storm tracks are moving poleward (Yin, 2005; Bengtsson et al., 2006; Ulbrich et al., 2008; Li et al., 2018). However, conflicting results regarding wind storm intensities have not allowed a clear understanding of expected changes in the evolution of extratropical wind storms. A decreasing trend in storminess indices has

been found in observations (Smits et al., 2005; Wever, 2012), consistent with observed large-scale near-surface wind decreases found over continental areas (Vautard et al., 2010; McVicar et al., 2012). By contrast, a more zonal flow is expected from climate projections (Haarsma et al., 2013), inducing a mean large-scale circulation more favourable to winter wind storms. Over the middle of northern Europe, along the track of highest mean wind speeds, a slight increase in extreme wind speeds was found in several model studies (Ulbrich et al., 2009; Mölter et al., 2016; Vautard et al.,

2014), while no consistent changes were found in wind storm number or intensities over the Mediterranean areas (Nissen et al., 2014). The frequency of occurrence of "sting jets", sometimes found in the strongest wind storms in the north-east Atlantic, has been suggested to be increasing, from climate model simulations (Martínez-Alvarado et al., 2018), but the area of concern is mostly over ocean.

Attribution of extreme weather events, an emerging scientific area (Stott et al., 2016), attempts to study changes that occurred for certain classes of events with specific magnitude, spatial scale, and timescale. The link between such events and climate change is often questioned by the media and the public when they occur; even though it may not yet be mature enough for these purposes, event attribution can also potentially be used for responsibility assessment when impacts and losses are present. While storm changes have been studied as a broad category, only few event attribution studies analysed the influence of human activities on such types of extratropical wind storms. There are few other studies on observed trends in wind storms over Europe and those results were mostly inconclusive. Vose et al. (2014) call trends over land "inconclusive", but find a trend over sea. Barredo (2010) finds no upward trend in losses indicating insignificant storminess change. Beniston (2007) finds a sharp decline in wind storms in Switzerland since about 1980 but a connection to the North Atlantic Oscillation (NAO) evolution is proposed.

In this article we take as an example two of the devastating wind storms that occurred during January 2018 in western Europe and analyse, using event attribution techniques, how the frequency of such storms has been and will be altered by human activities. For the first time we analyse both observations and high-resolution regional climate projections for our analysis, which are shown to fairly well simulate extreme wind speeds that are present in such storms.

In Sect. 2, we describe the meteorological context of the stormy month of 2018 in western Europe, the events studied, and their impacts. In Sect. 3, a quantitative characterization of the events is provided based on the analysis of observations. In Sect. 4, models and observations used are described. Sections 5 and 6 develop the analysis of each of the two storms analysed and Sect. 7 provides a summary and synthesis of the findings.

## 2 The stormy month of January 2018 and the studied storm cases

The year 2018 started with a series of four strong wind storms over western Europe. In particular, two major events pounded the continent: one on 3 January, named storm Eleanor by the Irish Meteorological Service (Met Éireann), and another one on 18 January, named storm Friederike by the Berlin Institut für Meteorologie.

Storm Friederike led to at least 11 casualties and major disruptions in the Netherlands and parts of Germany. In advance of storm Friederike, warnings were issued in both the Netherlands and Germany for severe wind gusts. On 18 January, the timing of the strongest winds was around 09:00–11:00 LT (local time, CET+1) just after the peak of the morning commute, with many people already on the road and in some cases caught by the strong winds. In addition to the wind hazard, snow created icy road conditions, causing car accidents with casualties. In Germany, according to the Insurance Journal (2018), storm Friederike is estimated to have caused around EUR 1.6 billion worth of damage. The authors estimate that this was the second most expensive storm to strike Germany in the past 20 years. In the Netherlands, three people were killed during the storm. For the first time in history, train traffic was completely shut down across the country. Amsterdam Airport Schiphol was closed and more than 300 flights were cancelled. Numerous roads were blocked by fallen trees and overturned trucks. Due to their height, trucks were susceptible to being blown off the roads, which caused disruptions and accidents.

The other major storm, storm Eleanor, led to major disruptions in France during the ski holiday season and is estimated to have cost as much as EUR 700 million (Insurance Journal, 2018). Ski resorts were closed for one or two days in the Alps, with significant economic consequences. Wind gusts of more than $130\,\mathrm{km\,h^{-1}}$ and nearing $150\,\mathrm{km\,h^{-1}}$ were reported over several flat regions in France and Switzerland. Large waves on the Atlantic coasts of Spain and France killed two people. Over France, according to the severity index developed by Météo-France, Eléanor was the sixth most severe storm since 1995.

The strongest wind gusts estimated from the European Centre for Medium-Range Weather Forecasts (ECMWF) are shown in Fig. 1 for both storms, which have a very different pattern.

More storms than these two were reported during January 2018. For instance, storm Carmen, which preceded storm Eleanor by two days, crossed southern France with wind gusts exceeding $130\,\mathrm{km\,h^{-1}}$. On 17 January, another storm, Fionn, passed over parts of the Mediterranean region and broke wind speed records, such as at Cap Corse, the northern edge of Corsica (winds reached $225\,\mathrm{km\,h^{-1}}$). In terms of number of events, January 2018 is the stormiest month since 1998 in France.

This exceptional storm activity was due to a strong westerly flow that persisted throughout the month (as shown in Fig. 2, first row) was enhanced by the jet stream extension eastward of its normal position. The persistence of the flow is also characterized by the frequency of occurrence of the so-called "zonal weather regime" (ZO), as defined by Michelangeli et al. (1995) using cluster analysis on sea level pressure (SLP) data from the NCAR/NCEP reanalysis. Approximately 45 % of the January days were classified in this cluster (Fig. 2, remaining panels), which is characterized by mild

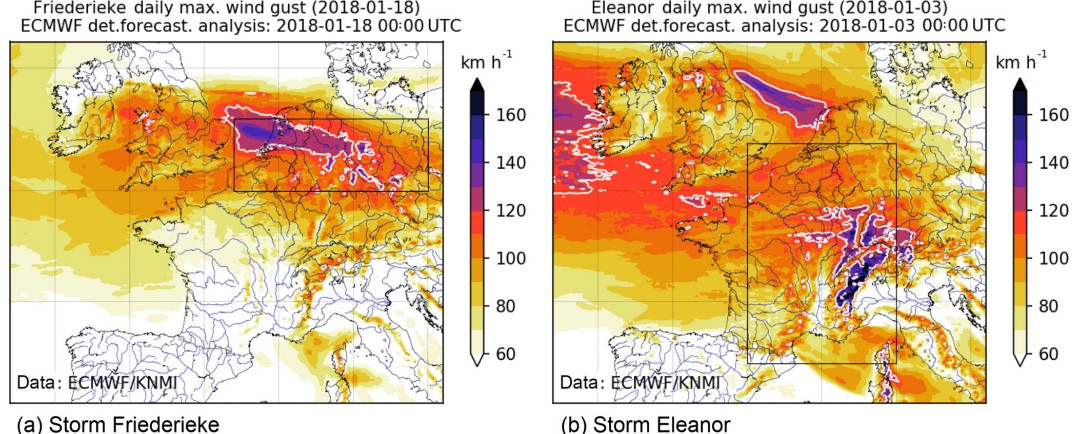

**Figure 1.** Strongest wind gusts during the storms Friederike **(a)** and Eleanor **(b)** as estimated from the ECMWF deterministic forecasts starting at 18 January 2018 00:00 UTC and 3 January 2018 00:00 UTC respectively. White contours are used to indicate areas where gusts exceed $118\,\mathrm{km\,h^{-1}}$. The boxes indicate the spatial event definitions (Sect. 3).

and wet winter weather. The average frequency of the ZO weather regime is close to 25 %. Although not exceptional, this high frequency is significantly higher than normal.

## 3 Event definitions

Classical event attribution relies on defining an event as an exceedance of a threshold in the tail of the distribution of an event indicator. Once the indicator is defined the probability of exceeding the threshold is calculated for the current climate and for a hypothetical climate where anthropogenic influence is not present or largely reduced. Once this is done, the ratio of the probabilities (probability ratio denoted hereafter as "PR") is estimated. It indicates how much more likely (PR > 1) or unlikely (PR < 1) the event is between the two climates. We define in this section the indicators associated with the two studied storms.

Storm Friederike was the result of rapidly developing cyclogenesis and the area with highest wind speeds, located south of the trough centre, moved rapidly from west to east. It crossed the Netherlands and central Germany in about half a day. In this analysis, the salient event characteristics will be represented by an indicator defined on the basis of daily maximum wind speed, derived from observations available from the Integrated Surface Database ("Lite" version, ISD-Lite; Smith et al., 2011). The database contains global hourly weather data for eight variables. Many of these observations are made at airports from cup anemometers. However, many stations only contain three-hourly data for the earlier part of the record. Also, when analysing outputs from some of the models contributing to EURO-CORDEX, the daily maximum near-surface wind speed was obtained on the basis of three-hourly wind speeds. For these reasons, we only sampled observations every 3 h and the daily maximum wind

speed was calculated only if at least four of the eight sampled observations were available.

In Fig. 3, we plot the values of the daily maximum wind observed over northwestern Europe on the days of the storms. The track of storm Friederike (Fig. 3a) can be seen in the box (50–53° N, 2–15° E) where wind speeds are largest. We therefore selected the seasonal (December–January–February, DJF) maximum value of this land area average of daily maximum wind speed as the event indicator (see also Fig. 1a).

This area contains 68 stations observing wind speed. The area average cannot be exactly calculated using the stations because the distribution of the stations is not even or dense enough, but we take the station average as a reasonable approximation. Using this indicator, storm Friederike is the eleventh strongest storm in the area since 1 January 1976, with an indicator value of $16.0\,\mathrm{m\,s^{-1}}$ max daily wind. The 2017–2018 winter season (DJF) becomes the seventh strongest in terms of strongest winter winds over this station network. However, this storm was not the seventh strongest storm as some seasons had multiple stronger storms. We also considered the daily mean wind for models that did not store higher-frequency data. In terms of that indicator, Friederike was not remarkable with $8.7\,\mathrm{m\,s^{-1}}$, as it was a very short duration storm with a calm period immediately following it, bringing the daily mean to a moderate value.

For models, the area average is calculated over land grid points, which slightly lowers the indicator value (see comparisons in Table 1 for model evaluation) as the stations are concentrated near the coast where the intensity was higher. In order to calculate seasonal return periods, we take the maximum value of the indicator over the winter season (DJF).

The structure of storm Eleanor was very different. Eleanor was embedded in a deep large-scale low-pressure system. Its strong winds affected a much broader area than storm

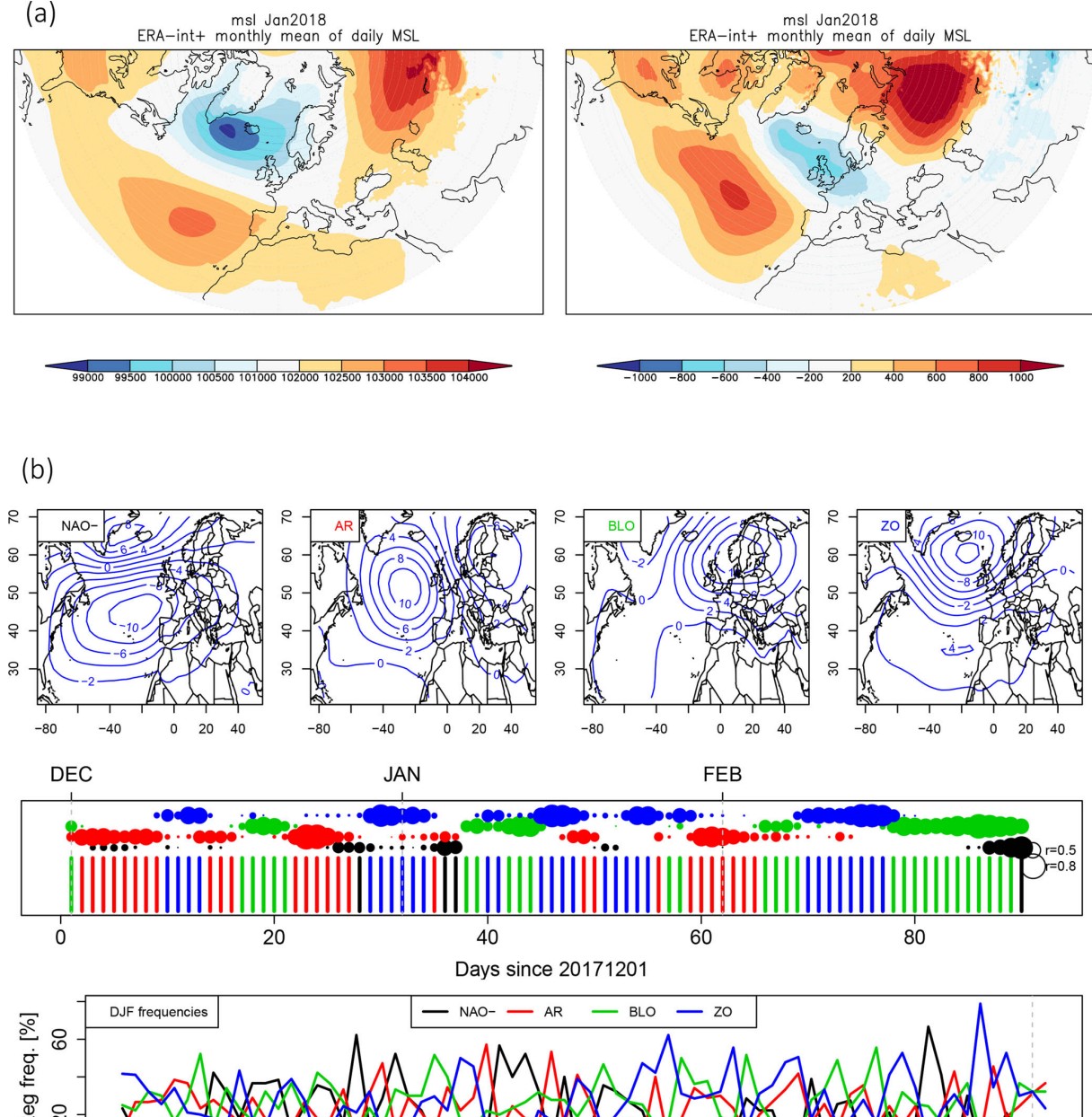

**Figure 2. (a)** Sea level pressure (left) and anomaly (right) (ECMWF analysis, ERA-Interim climatology); **(b)** first row: weather regime cluster centroids from NCAR/NCEP reanalysis; second row: occurrence of weather regimes from 1 December 2017 to 28 February 2018; the vertical bars indicate the preferred centroid (NAO-; Atlantic Ridge, AR; Scandinavian Blocking, BLO; Zonal, ZO) and the coloured circles indicate the spatial correlation with the preferred centroid; third row: weather regime wintertime frequencies from 1948 to 2018.

Friederike: from Ireland and the UK via western France to Switzerland and the Riviera coast. Its high wind speeds, unusual in western Europe, constituted its most striking aspect. As this storm also passed within a day, we construct the same indicators as for Friederike, which are daily maximum and

mean of wind speed, but averaged over a much wider area, from 42 to 52° N and 0 to 10° E (see Figs. 1b and 3b). The value of the indicator is 12.3 m s$^{-1}$ for maximum winds and 8.3 m s$^{-1}$ for daily mean winds.

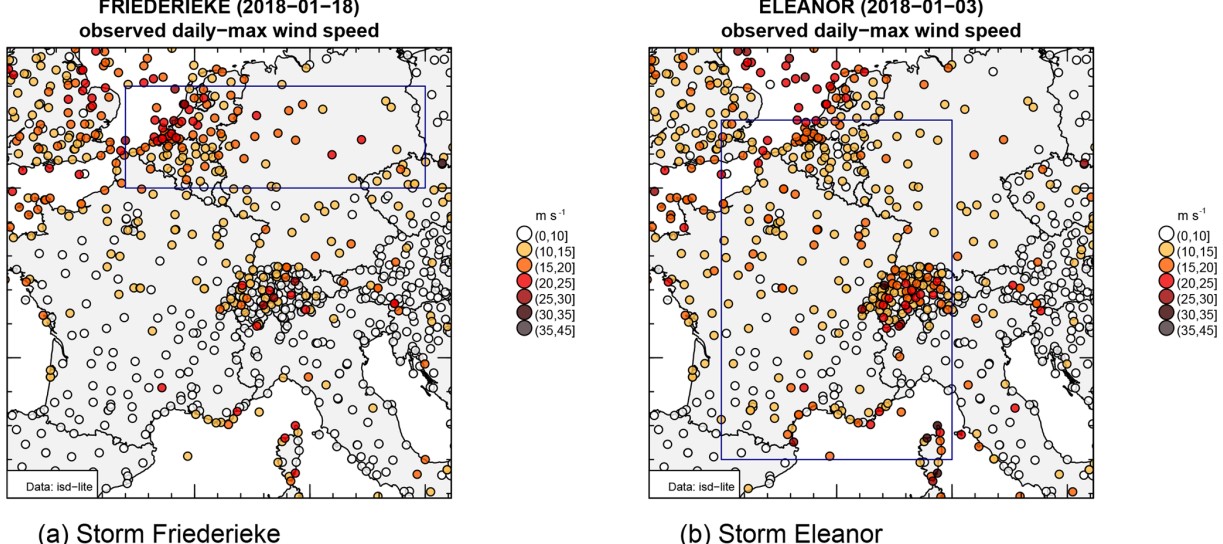

**Figure 3.** Daily maximum wind speeds at ISD-Lite stations over northwestern Europe and area defining the event indicator for storms **(a)** Friederike and **(b)** Eleanor.

**Table 1.** Mean (WXm), 95th (WX95), and 99th (WX99) percentiles of the distribution of the daily maxima of wind speed averaged over the 68 stations or the land grid points (for models) for the winter season. Italics indicate that the daily mean rather than the daily maxima of the wind speed statistic has been used. Observation cells (Obs.) are the first two data rows, the upper one gives the daily maximum of the wind speed statistic, and the lower one (in italics) gives the daily mean of the wind speed statistic. For the three model ensembles (RACMO, EURO-CORDEX (pooled), and HadGEM3-A), both station (station) and area (area) averages are shown. For weather@home only the area average was available.

| Model/ensemble (values in metres per second) | WXm | WX95 | WX99 | WXm | WX95 | WX99 |
|---|---|---|---|---|---|---|
| | | Friederike | | | Eleanor | |
| Obs. (ISD-Lite) daily max[2] | 6.4 | 11.4 | 13.5 | 6.1 [5.8] | 9.7 [9.5] | 11.1 [11.0] |
| Daily mean | *4.5* | *8.4* | *10.1* | *4.1* | *6.6* | *7.7* |
| RACMO (16 members) (station) | 6.8 | 10.9 | 12.6 | 5.3 | 8.2 | 9.4 |
| RACMO (16 members) (area) | 6.6 | 10.4 | 12.0 | 5.5 | 8.5 | 9.8 |
| HADGEM3-A[1] (15 members) (station) | *4.4* | *8.1* | *9.6* | *3.5* | *5.7* | *6.7* |
| HADGEM3-A[1] (15 members) (Area) | *3.9* | *7.6* | *9.0* | *3.1* | *5.4* | *6.3* |
| bc-EURO-CORDEX (pooled) (station) | 5.9 | 10.2 | 12.0 | 4.7 | 7.5 | 8.7 |
| bc-EURO-CORDEX (pooled) (area) | 5.6 | 9.7 | 11.5 | 4.7 | 7.8 | 9.1 |
| bc-ARPEGE (zoomed version) | 5.7 | 9.8 | 11.9 | 4.8 | 7.6 | 8.9 |
| bc-RACMO+HADGEM | 6.2 | 10.4 | 12.0 | 4.9 | 7.9 | 9.0 |
| bc-RACMO+EC-EARTH | 6.2 | 10.4 | 12.2 | 4.7 | 7.8 | 9.2 |
| bc-REMO+MPI | 6.0 | 10.2 | 12.1 | 4.7 | 7.8 | 9.0 |
| bc-WRF+IPSL | 5.9 | 10.2 | 12.1 | 4.7 | 7.9 | 9.0 |
| bc-HIRHAM+EC-EARTH | 5.8 | 10.2 | 11.8 | 4.7 | 7.8 | 9.2 |
| bc-RCA+ARPEGE | 5.8 | 10.0 | 11.8 | 4.7 | 7.9 | 9.2 |
| bc-RCA+IPSL | 5.8 | 10.0 | 11.7 | 4.7 | 7.8 | 9.2 |
| bc-RCA+HADGEM | 5.7 | 10.1 | 11.6 | 4.7 | 7.9 | 9.2 |
| bc-RCA+MPI | 5.9 | 10.0 | 11.9 | 4.8 | 8.1 | 9.1 |
| bc-RCA+EC-EARTH | 5.6 | 10.0 | 11.8 | 4.5 | 7.8 | 9.0 |
| *weather@home*[1] TS1 | *6.4* | *11* | *11.8* | *5.22* | *8.45* | *9.03* |

[1] Only daily mean winds available, so statistics only from daily means. [2] For Eleanor, averages made with stations north of 43.5° N are in square brackets.

## 4   Observations, model ensembles, and evaluation

For the observational part of the attribution analysis, we used two sources of station data. Unfortunately, the available quantities were slightly different in the different datasets. The analysis is mainly based on the ISD-Lite database described above, in which we used the daily maximum of three-hourly instantaneous wind speed. Additional results are based on the KNMI climatological service database, which provides the daily maximum of the hourly averaged wind speed at 34 weather stations in the Netherlands. The highest hourly wind of the year series were visually quality controlled. For three series, early data were discarded for obvious inhomogeneities supported by the metadata (Leeuwarden before 1990; De Bilt before 2002; Lichteiland Goeree before 1995). Most series start in 1981, but they are notably more variable and possibly unreliable before circa 1990.

The KNMI data at these stations plus 22 sea stations were also converted to potential winds, i.e. the wind speed at 10 m that would have occurred assuming a roughness length of 3 cm over land and 2 mm over water, and assuming neutral stability (Wever and Groen, 2009). Such a calculation is made by multiplying wind speeds by "exposure correction factors" which to first order account for changes in the elevation of the wind anemometer and changes in roughness surrounding the station in different directions. These factors are deduced from the high-frequency variability in the wind (intra 10 min standard deviation or wind gust). These exposure correction factors are recomputed every 3 years. Three years of measurements are required to ensure that there are at least 10 appropriate measurements in each of the wind direction sectors. If a new exposure correction factor is found to be significantly different (absolute difference > 0.05) from the existing factor the new factor is introduced.

We used four complementary ensembles of climate model simulations. Two of them are made of regional climate simulations by downscaling low-resolution global climate models (GCMs) with a high resolution (12.5 km). One of these is using the same model chain with different members for the GCM, while the second one is a multimodel ensemble member. We therefore cover several aspects of the uncertainty. The other two ensembles were available at the time of the study and also used, one of which consists of a very large ensemble. However, for these latter ensembles only daily mean wind speed was available while for the former daily maximum wind speed was available. Our assessment is therefore rather based on the first two ensembles, which better represent the January 2018 storms, the other ensembles being used for consistency checking.

The first ensemble is the RACMO regional climate model ensemble downscaling 16 initial-condition realizations of the EC-EARTH 2.3 coupled climate model in the CMIP5 RCP8.5 scenario (Lenderink et al., 2014; Aalbers et al., 2017). The RACMO model uses a 0.11° (12 km) resolution and the daily maximum of near-surface wind speed is

analysed. In RACMO, the near-surface wind speed is diagnosed from the model wind and stability vertical profile as the wind speed at 10 m, applying a roughness length of at most 3 cm for land grid points, and a Charnock-type relation for sea grid points (van Meijgaard et al., 2008). This ensemble was previously used to estimate the change in the odds of wind stagnations in northwestern Europe (Vautard et al., 2017) and was found to simulate monthly wintertime wind speeds over western Europe in a satisfactory manner. RACMO simulations are available for the 1950–2100 period. As in previous analyses (e.g. Philip et al., 2018), we use a 20th century early 30-year period (1951–1980) to estimate odds in the past climate, and the 2001–2030 period to estimate odds in the current climate. We also use two future periods, a period called "near future" (2021–2050) and a period called "mid-century" (1941–1970). We only used the simulations using the RCP8.5 radiative forcing scenario. As a cross-check we fitted a time-dependent generalized extreme value (GEV) function to the whole period 1971–2070, as described in van der Wiel et al. (2017).

The second ensemble is the multimodel EURO-CORDEX ensemble (Jacob et al., 2014), using a 0.11° resolution over Europe. For this ensemble, only 11 simulations were used and a bias correction was applied (Bartok et al., 2018 TS2) using the cumulative distribution function transform (CDFt; Vrac et al., 2016). These simulations have been evaluated in the context of the CLIM4ENERGY Copernicus Climate Change Service project (http://clim4energy.climate.copernicus.eu, last access: 25 July 2018). The reference data used for bias correction is the Watch Forcing Data ERA-Interim (WFDEI; Weedon et al., 2014). For wind speed, it is essentially an interpolation of ERA-Interim over a $0.5° \times 0.5°$ grid. This dataset has a relatively low resolution, so extreme winds are not expected to be accurately represented. This weakness is, therefore, probably propagated to the EURO-CORDEX ensemble. The ensemble is pooled, which is formally possible because the bias correction method corrects data making it homogeneous across the multimodel distribution.

The third model ensemble is the HadGEM3-A ensemble (Ciavarella et al., 2018; Vautard et al., 2018), which includes a set of 15 realizations of atmospheric simulations using observed sea surface temperatures (SSTs; reflecting the actual world) and a set using SSTs where the CMIP5 mean patterns of anthropogenic heat contribution are removed to estimate the ocean response to a pre-industrial atmospheric composition (as the natural/counterfactual world). The latter runs also use pre-industrial greenhouse gas and aerosol concentrations. Land use, and hence roughness, is set to 1850 values in the counterfactual ensemble. For this model, the wind speed daily maximum was not available and the daily mean wind was used instead. No future simulations were available.

The fourth ensemble is obtained from simulations using the distributed computing framework known as weather@home (Massey et al., 2015). We used four different

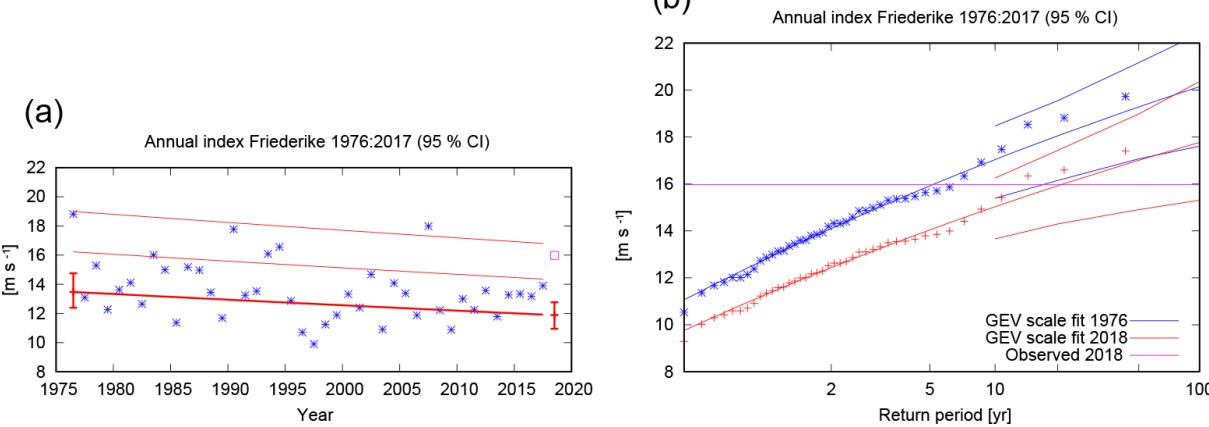

**Figure 4. (a)** Highest winter value of the Friederike index described in Sect. 2 fitted to a generalized extreme value (GEV) function that scales with time. The thick line denotes the position parameter $\mu$, the thin lines the 1 in 6 years and 1 in 40 years return values. **(b)** The GEV fit as a function of return period for the climate of 1976 (blue, observations have been scaled up with the fitted trend) and 2018 (red).

large ensembles of December–February wind speeds using the Met Office Hadley Centre for Climate Science and Services regional climate model HadRM3P at 25 km resolution over Europe embedded in the atmosphere-only global circulation model HadAM3P at N96 resolution. The first set of ensembles represents possible winter weather under current climate conditions. This ensemble is called the "all forcings" scenario and includes human-caused climate change. The second set of ensembles represents possible winter weather in a world as it might have been without anthropogenic climate drivers, using different estimates of pre-industrial SST deduced from the CMIP5 ensemble and pre-industrial greenhouse gas and aerosol concentrations. Land-use in both ensembles is identical. This ensemble is called the "natural" or "counterfactual" scenario (Schaller et al., 2016). The third set of ensembles represents a future scenario in which the global mean surface temperature is 1.5 °C higher than pre-industrial global temperatures. The fourth scenario is the same as the third, but for 2 °C of future global mean temperature anomaly. To simulate the third and fourth scenarios, we use atmospheric forcings derived from RCP2.6 and RCP4.5 and sea surface temperatures that match the atmospheric forcing obtained from CMIP5 simulations (Mitchell et al., 2017).

The evaluation of the models' ability to simulate the indicator is made using the ISD-Lite observations, which are available in near-real time. In order to evaluate the capacity of the models to simulate the winds, we extracted wind speed daily maxima at the locations of ISD-Lite stations and averaged these values over all stations in the area. Then, we compared the simulated mean, 95th centile, and 99th centile with the observed equivalent for each model ensemble (Table 1). For HadGEM3-A and weather@home, as daily maxima were not available, we used daily averages of the wind speeds.

For RACMO, HadGEM3-A, and weather@home, model values are pooled together to compute the distribution statistics. For EURO-CORDEX, we calculated both individual model and pooled statistics. Results are presented in Table 1 for the average over all grid points closest to the 68 ISD-Lite stations, together with equivalent statistics when the average is made over all land grid points, instead of the positions of the stations. Results show that the models reproduce the indicator with success along the distribution. Comparisons to station data indicate a general underestimation of models within a 10 % range. EURO-CORDEX simulations are bias corrected, so the bias is essentially reflecting the WFDEI (ERA-Interim based) bias. The fact that statistics do not differ from one model to the other supports pooling the models' simulations together in a common distribution. This bias is consistent with models not simulating observational noise due to remaining turbulence. For weather@home, we only have daily values for mean wind speed, so we calculate the maximum mean area-averaged daily wind speed in a winter season. The simulated values are higher than the observed values for this quantity, especially for the mean, while the 95th and 99th percentiles are comparable to observations in particular for storm Friederike.

Grid point averages reach lower values than station averages, which is a probable consequence of the higher density of stations near the North Sea coast where winds are stronger, which is reflected in the observed area average. The factor between observation statistics and model statistics for station averages is rather uniform across the distributions, even though the distribution is more heavy-tailed for observations than for simulations. In order to homogenize attribution results among models and observations and compare return periods with observations, we scaled all simulations by the ratio between 99th centiles of observed station averages and simulated grid-point averages. These bias corrections are a

factor of 1.13 for RACMO (for both storms), 1.17 (1.28) for EURO-CORDEX for Friederike (Eleanor), and 1.12 (1.22) for HadGEM3-A for Friederike (Eleanor).

In each case, the attribution of the event consists of comparing probabilities of the exceedance of an indicator, e.g. the winter maximum of the daily maximum wind speed averaged over a specific area depending on the storm studied (see below for the definition for each case) in the current climate with the probability of the exceedance in a world where anthropogenic forcing on climate was not present or was weaker. For observations, this is done by fitting an extreme value distribution as observations are not numerous. In this case the parameters of the distribution are taken to be functions of global temperature, as in previous studies (e.g. Philip et al., 2018). Using model ensembles, this is done in a nonparametric way by pooling all winter maxima of each ensemble member into a single pool and computing the probability by counting the number of exceedances of the threshold. To obtain confidence intervals, this procedure is done within a bootstrap framework where random drawings are done including possible repetitions. The 95 % confidence intervals are obtained by taking the 5th and the 95th return periods of the bootstrap sample, in a procedure similar to Vautard et al. (2017), while the median value is used as a best estimate. Once a value is selected in the bootstrap, one forces the whole model series to be selected. The probability ratios (PRs) are calculated in the same way by calculating ratios of probabilities for each return value in a bootstrap framework. The reference in this case is taken as the "current climate" and ratios are calculated relative to this period. When the confidence interval of this ratio does not include 1, one concludes that the probability is significantly changing due to climate change. Results from observations and models are comparable in terms of climate time period as the parametric method used for observations accounts for time dependence.

## 5 Storm Friederike

### 5.1 Observations

As mentioned before, we compute the daily maximum of three-hourly wind speed (taken at 00:00, 03:00, 06:00, 09:00, 12:00, 15:00, 18:00, and 21:00 UTC) at each station and averaged it over the ISD-Lite stations available in the box. The winter maximum of this quantity is shown in Fig. 4a as a function of time (labelled with the year of the second half of winter). The data have been fitted by a GEV distribution in which the location parameter $\mu$ and scale parameter $\sigma$ vary exponentially with time, such that their ratio remains constant. The shape parameter $\xi$ is not time-dependent. This fit shows a significant decrease ($p < 0.05$ two-sided) in wind speed over 1976–2017, in agreement with earlier analyses (Smits et al., 2005; Vautard et al., 2010). The decrease in intensity of about 12 % with a 95 % confidence interval (CI) of 0 % to 30 %, which we will denote thereafter (95 % CI: 0 %

to 30 %), corresponds to a decrease in probability of about a factor of 4 (95 % CI: 1 to 100) (see Table 2). Using the global mean temperature as a covariate instead of time gives slightly higher trends. The shape parameter $\xi$ of the GEV is most likely negative, so the distribution has a tail that is thinner than an exponential distribution. This implies that the ratio of probabilities is larger for the same difference in wind speed for a higher baseline, i.e. stronger storms.

The return period of an event like Friederike or worse in the area in which the indicator value reached $16.0\,\mathrm{m\,s^{-1}}$ on 18 January in the current climate is 13.5 years as estimated directly from the data (Table 1). By looking at the GEV fits (Fig. 4), in the 1970s, this was roughly 5 years, so the event, defined using our indicator, has become a fairly rare event due to the decrease in high wind speeds observed during this period.

The result is confirmed in a different dataset from KNMI observations (not shown), with most stations showing a clear downward trend over the whole period (1971–2017 for two stations, 1982–2017 for most others). A simultaneous fit to all stations scaled to the same mean show a decrease in intensity of $-15\,\%$ (95 % CI: $-7\,\%$ to $-17\,\%$), the same as the ISD-Lite data show. The trends are much less clear when starting in 1990 (using stations with at least 25 years of data). The trends in potential wind are much smaller (around $-5\,\%$) and not significantly different from zero, even when pooled over all stations.

### 5.2 RACMO ensemble

The storm indicator is scaled to have the same 99th percentile as the observed indicator in the historical period. Indicator statistics are then obtained for four climate periods: 1951–1980, simulating the "past" period; the "current" period taken as 2001–2030; and two future periods assuming the RCP8.5 scenario (2021–2050 and 2041–2070). The observed indicator value for storm Friederike ($16.0\,\mathrm{m\,s^{-1}}$) has a return period of about 13 years for the current climate (95 % CI 10–19 years, see Fig. 4 and Table 1), which is close to observed values. The probability of witnessing higher indicator values is no different or marginally significantly different between past and current periods (Fig. 5a and d). However, the change of probability becomes larger in future periods, with a probability ratio (PR) of about 1.5 (95 % CI: 1–2) in the near term (Fig. 5d and Table 2). For this particular case, the increase is also stronger for stronger storms due to an increase in the variability relative to the mean.

Therefore, according to this model's representation, we do not identify a climate change impact currently, but the increase in probability of storms like Friederike emerges in the coming decades. A fit with a GEV that scales with the global mean temperature of the driving EC-Earth ensemble gives no change (PR between 0.95 and 1.16), overlapping with the 30-year time window analysis (not shown).

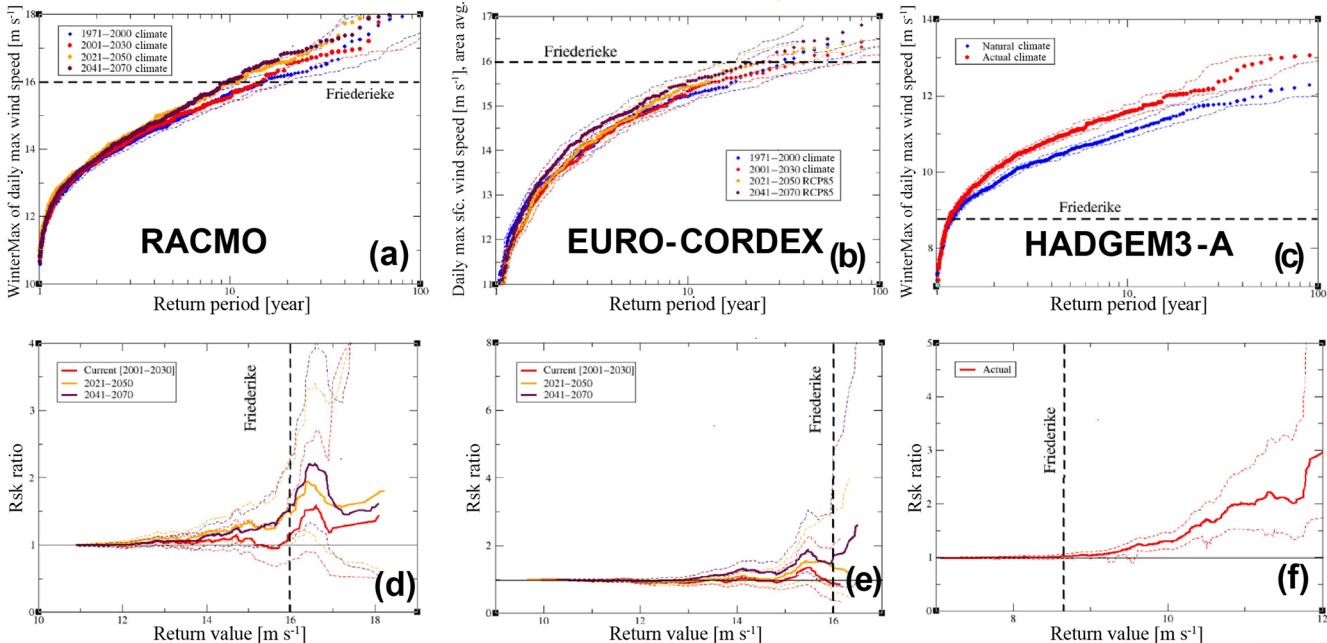

**Figure 5. (a–c)** Return values as a function of return periods for the storm Friederike indicator, for different time periods and the RACMO **(a)**, EURO-CORDEX **(b)**, and HadGEM3-A **(c)** ensembles. **(d–f)** Probability ratio of exceeding the return value of the indicator as compared with the counterfactual period as a function of the return value, with 5 %–95 % significance intervals as dashed lines with corresponding colour, calculated from a nonparametric bootstrap and shown for RACMO **(d)**, EURO-CORDEX **(e)** and HadGEM3-A **(f)** ensembles.

**Table 2.** TS3 Event return periods and probability ratios summarized for all model ensembles and for storm Friederike. Probability ratios (PRs) are calculated with respect to a past or counterfactual period. PI is pre-industrial and n/a CE1 is not applicable.

| Ensemble | Ret. period (year) | PR for current climate | PR for period 2021–2050 | PR for period 2041–2070 | PR for period PI +1.5 °C | PR for period PI +2.0 °C |
|---|---|---|---|---|---|---|
| Obs. ISD-Lite | | | n/a | n/a | n/a | n/a |
| Models using wind speed daily maximum (over three-hourly data) | | | | | | |
| RACMO | 15 [11–20] | 1.1 [0.8–1.7] | 1.5 [1.0–2.2] | 1.5 [1.1–2.3] | – | – |
| EURO-CORDEX | 40 [25–80] | 0.9 [0.4–2.0] | 1.4 [0.7–3.0] | 1.6 [0.8–4.1] | – | – |
| Models using wind speed daily mean | | | | | | |
| HadGEM3-A | 1.2 [1.15–1.27] | 1.02 [0.98–1.06] | – | – | – | – |
| weather@home | 1.3 [1.29–1.35] | 1.03 [0.97–0.2] TS4 | – | – | 1.039 [0.98–1.16] | 1.04 [0.98–1.17] |

## 5.3 EURO-CORDEX ensemble

In the EURO-CORDEX simulations, the return period corresponding to the scaled indicator (25–40 years) is larger, making it a more extreme event (Fig. 5b). The shape of the distribution is clearly different from that of the RACMO simulations and that of the observations (compare with Fig. 5a and b). However, as for RACMO, the PR (Fig. 5e) is not significantly different from 1 (see Fig. 5e and the low boundary of the 95 % confidence interval), despite systematic values of PR above 1. Such an increase becomes marginally significant in the middle of the century with PR values in the range 1 to 3 for lower wind thresholds. Again, this indicates a ten-

dency for more storms like Friederike in the future with an anthropogenic signal emergence not yet achieved. The GEV with smoothed EC-Earth global mean temperature as covariate confirms this conclusion, with an increase in probability of 1.0 to 1.2 ($p \sim 0.1$); this corresponds to the assumption that the percentage increase is a constant 0.0 % to 1.4 % per degree global warming over the whole range of Fig. 5a–b. This assumption holds well over the four 30-year time periods considered before.

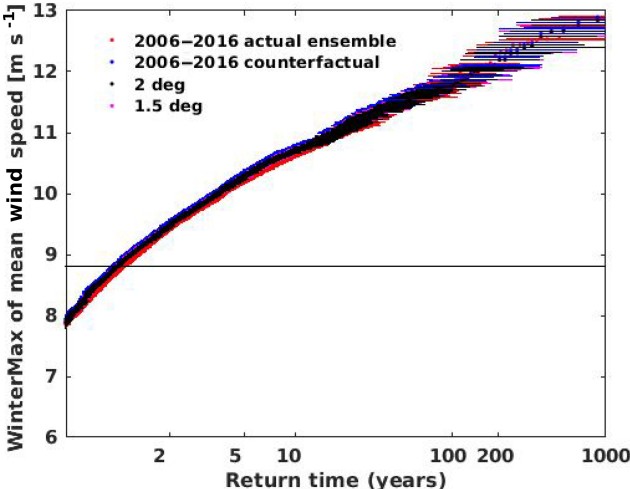

**Figure 6.** Return values as a function of return periods for the storm Friederike indicator, for the weather@home ensemble with 5 %–95 % significance intervals, calculated from a nonparametric bootstrap, under the assumptions of actual, counterfactual, 1.5 °C, and 2 °C warming.

## 5.4 HadGEM3-A ensemble

The HadGEM3-A ensemble exhibits a significant difference between actual and counterfactual periods, with a current increase in strong daily mean winds in the area struck by storm Friederike (Fig. 5c). However, due to the use of the mean wind speed instead of the maximum wind speed, the indicator does not disentangle extreme winds over a short time period from less strong winds over an extended time period (Fig. 5c). Accordingly, the observed value is not exceptional, due to the fast travelling nature of the extremely high winds in the area: for the value corresponding to Friederike ($8.7 \, \mathrm{m \, s^{-1}}$), such events occur almost every year in both types of simulations.

## 5.5 weather@home

For weather@home, using the suboptimal definition of maximum of daily mean wind to define storm Friederike ($8.7 \, \mathrm{m \, s^{-1}}$), we find no significant change in the likelihood of storms like Friederike to occur (Fig. 6). In contrast to the EURO-CORDEX assessment, this also holds for rarer events (not shown).

## 6 Storm Eleanor

Results for storm Eleanor confirm the findings obtained for storm Friederike. All numerical results are presented in Table 3.

## 6.1 Observations

The same observational analysis on the Eleanor index as in Sect. 4.1 gives a more significant downward trend for this storm ($p < 0.01$ two-sided), with a decrease of about 20 % (3 %–35 %) (Fig. 7). This corresponds to an increase in return period of a factor of 8 (95 % CI 1.5–100). The return period is also about 20 years in the current climate according to the fit of Fig. 7.

## 6.2 RACMO ensemble

Storm Eleanor is now investigated through the wind daily maximum indicator with an average over the large region as defined above. Due to the southern boundary that is excluding a small band of the large region, we used a boundary at 43.5° N instead of 42° N for this model. This makes the indicator return value for stations slightly lower than when calculated over the full region ($11.9 \, \mathrm{m \, s^{-1}}$ instead of $12.3 \, \mathrm{m \, s^{-1}}$). The corresponding RACMO return period is in the range of 3 to 5 years. The climate change is not significant for the current period and marginally significant for future periods, as for storm Friederike (Fig. 8a, d). The estimated PR is 1.1 and slightly higher for future periods. Interestingly, for stronger storms, the PR increases. The same results hold for a GEV fit with covariate of all data in 1971–2070, with a PR significantly different from 1 (95 % CI 1.0 to 1.2, not shown).

## 6.3 EURO-CORDEX ensemble

Using the EURO-CORDEX ensemble, the return period of the large-scale storm Eleanor, characterized by the chosen indicator, is estimated to about 7–10 years (Fig. 8b). A climate change signal is absent in the simulations when comparing periods 1971–2000 and 2001–2030. For the indicator value, the PR is in the range [0.5–1]. Only for later periods and for larger indicator values, a marginally significant increase in the PR in the range [1–2] can be seen (Fig. 8b, e). A GEV with a modelled global mean temperature (from EC-Earth) as covariate also gives a nonsignificant increase with a PR between 0.99 and 1.15 (95 % CI).

## 6.4 HadGEM3-A ensemble

For HadGEM3-A, using the daily mean wind, we find no climate change signal in the estimation of the probabilities of high winds of any magnitude; but for the very extreme winds, we find marginally significant changes in the direction of more frequent high winds under current conditions than under natural conditions (Fig. 8c, f). The estimated return period for the indicator value corresponding to Eleanor, which does not fall in the extreme tail, also lies between 3 and 5 years.

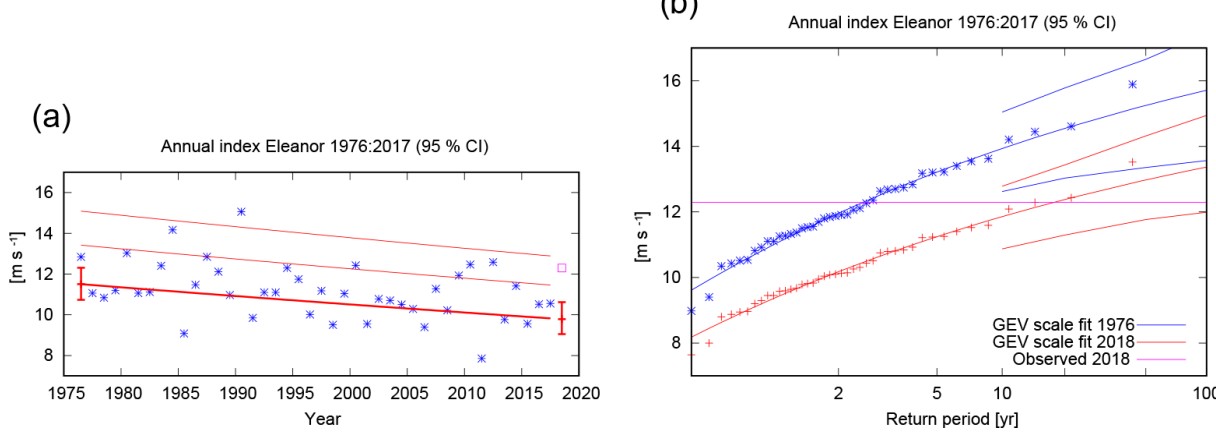

**Figure 7.** Same as Fig. 4 but for the Eleanor index.

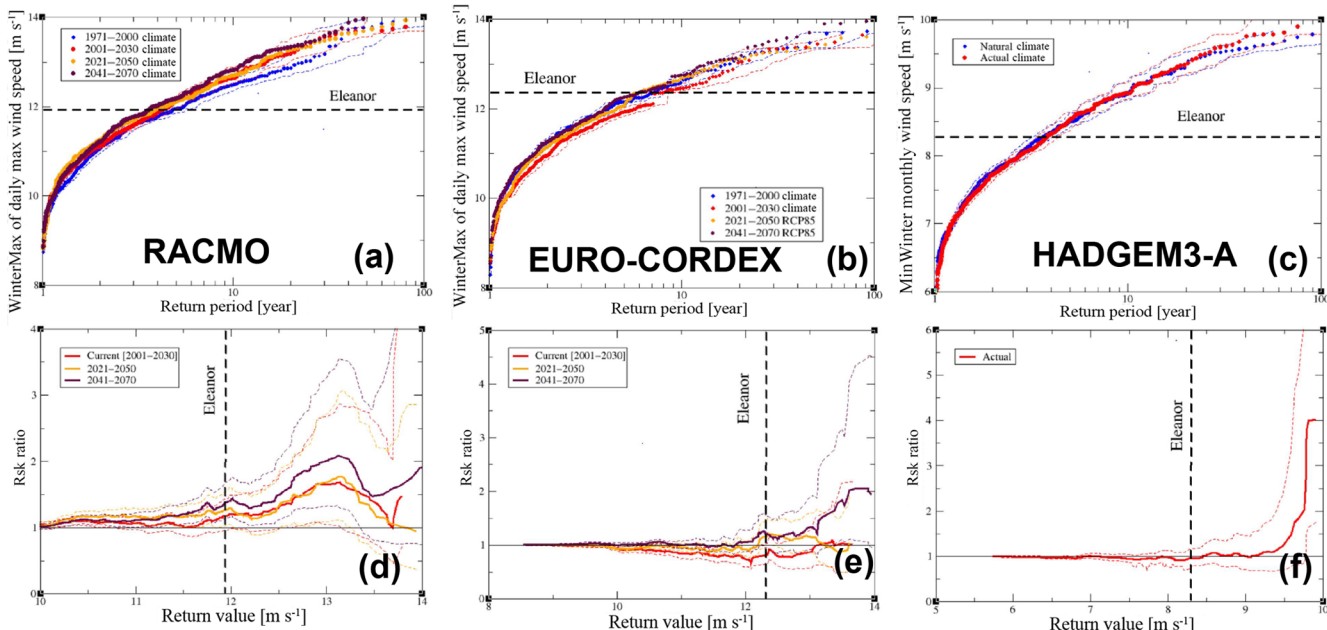

**Figure 8.** Same as Fig. 5 for storm Eleanor.

## 6.5   weather@home

For weather@home, using the maximum of daily mean wind to define storm Eleanor (8.3 m s$^{-1}$), we find no significant change in the likelihood of storms like Eleanor to occur (Fig. 9). In contrast to the EURO-CORDEX assessment, this also holds for rarer events where the weather@home model shows a nonsignificant decrease in high wind speeds.

## 7   Sensitivity to the domain definition

Domain definitions have been taken from a visual inspection of areas of maximal wind speeds. However, the choice bears some arbitrariness. In order to test the robustness of our re-

sults to changes in the domain definition, we have recalculated the changes in return periods of the same indicator calculated over a smaller domain and a larger domain for each storm (see Table 4 for the domain coordinates). This calculation is made only with the RACMO simulations. Results for each domain are reported in Table 4. Each row indicates the return period of an event with the same current-climate return period as the storm for each domain. Changing domain induces a change in return value for a given return period. Confidence intervals are indicated. As shown in the table, the return period almost systematically decreases with time, but 95 % significance is generally not reached for the difference between current period and past period. Our result is therefore robust to a change in the domain definition.

**Table 3.** Event return periods and probability ratios summarized for all model ensembles and for storm Eleanor. PI is pre-industrial and n/a is not applicable.

| Ensemble | Ret. period (year) | PR for current climate | PR for period 2021–2050 | PR for period 2041–2070 | PR for period PI +1.5 °C | PR for period PI +2.0 °C |
|---|---|---|---|---|---|---|
| Obs. ISD-Lite | | | | | n/a | n/a |
| RACMO | 4.2 [3.7–4.8] | 1.2 [1.0–1.4] | 1.3 [1.0–1.5] | 1.3 [1.1–1.6] | – | – |
| HadGEM3-A | 3.9 [3.4–4.5] | 1.0 [0.8–1.2] | – | – | – | – |
| EURO-CORDEX | 6.6 [5.6–6.9] | 0.8 [0.6–1.0] | 1.2 [1.0–1.4] | 1.2 [0.9–1.6] | – | – |
| weather@home | 13.9 [13.6–15] | 1.01 [0.62–2.35] | – | – | 0.94 [0.6–2.35] | 0.94 [0.59–2.36] |

**Table 4.** Sensitivity experiment results. Return periods and their confidence intervals (in square brackets) for the different time periods and the RACMO simulations. Each column contains results for one domain experiment. Return periods are all given for return value characteristics of the storms (15 years for Friederike and 4.2 years for Eleanor); F0 is storm Friederike, domain as in Table 2; F1 is the same with a smaller domain focused more on the Netherlands; F2 is larger domain including parts of UK; E0 is storm Eleanor, as in Table 3; E1 is the smaller domain; E2 is the larger domain. The coordinates (Lat, Long) of domains are indicated in the first data row.

| Domain | F0 | F1 | F2 | E0 | E1 | E2 |
|---|---|---|---|---|---|---|
| Coord. (Lat, Long) | 50–53, 2–15 | 50.5–52.5, 3–7 | 49–54, 0–18 | 43.5–52, 0–10 | 44–48, 3–9 | 43.5–53, −5–10 |
| RP 1951–1980 | 16.0 [12–22] | 11.2 [9–15] | 20.0 [15–28] | 4.8 [4.2–5.6] | 5.1 [4.5–5.9] | 4.6 [4.0–5.3] |
| RP 2001–2030 | 15.0 [11–20] | 15.5 [12–21] | 15.0 [11–20] | **4.2 [3.7–4.8]** | **4.2 [3.7–4.8]** | 4.2 [3.7–4.8] |
| RP 2021–2050 | **11.2 [9–14]** | 9.4 [8–12] | **12.3 [10–16]** | **3.9 [3.4–4.4]** | **4.1 [3.6–4.8]** | **3.9 [3.4–4.4]** |
| RP 2041–2070 | **10.2 [8–13]** | 9.6 [8-12] | **10.4 [8–13]** | **3.6 [3.2–4.1]** | **3.6 [3.2–4.1]** | **3.8 [3.3–4.3]** |

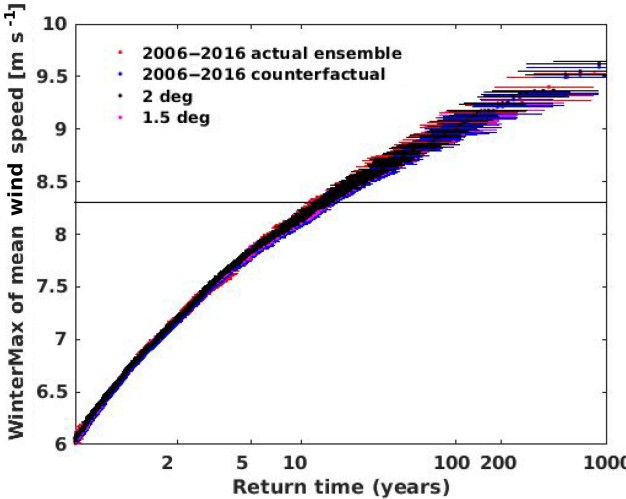

**Figure 9.** Return values as a function of return periods for the storm Eleanor indicator for the weather@home ensemble with 5 %–95 % significance intervals, calculated from a nonparametric bootstrap, under the assumptions of actual, counterfactual, 1.5 °C, and 2 °C warming.

## 8 Synthesis and conclusions

Western European countries have been struck by high-impact wind storms during the month of January 2018. The link between storms like Eleanor (on 3 January 2018) and Friederike (on 18 January 2018) and human-induced climate change

have been studied through this attribution analysis involving several simulation ensembles and observations from tens of weather stations.

From an analysis of two sets of observations, we conclude that near-surface storms in the areas of the two storms have a decreasing trend in wind speed and, hence, in frequency over the past 40 years, consistent with previous observation-based studies on storminess in these areas (Smits et al., 2005; Soubeyroux et al., 2017) and with global land wind stilling (Vautard et al., 2010; McVicar et al., 2012). This trend was shown to be close to zero over the Netherlands area when using the potential wind, indicating a strong influence of roughness changes there, as also demonstrated by Wever (2012). Other processes, such as aerosol increase, could also induce a wind decrease (Bichet et al., 2012), and decadal-scale long-term variability has been shown to have a significant role as well (e.g. Matulla et al., 2008).

We next turn to the model results. Due to the differing experiments that we used, the probability ratios have been computed over different intervals. To compare those we need to convert them to a common interval. We do this by assuming the probability ratio is an exponential function of some indicator of global warming $f(\text{yr})$:

$$\text{PR}(y_1, y_{\text{end}}) = \text{PR}(y_2, \text{yr}_{\text{end}})\text{PR}(\text{yr}_1, \text{yr}_2)$$
$$= \text{PR}(\text{yr}_2, \text{yr}_{\text{end}})\exp[f(\text{yr}_1) - f(\text{yr}_2)]. \quad (1)$$

We prefer not to use time for $f(\text{yr})$, as the warming trend is not linear in time. The global mean temperature as a proxy for the trend in the local temperature is often used (e.g. van

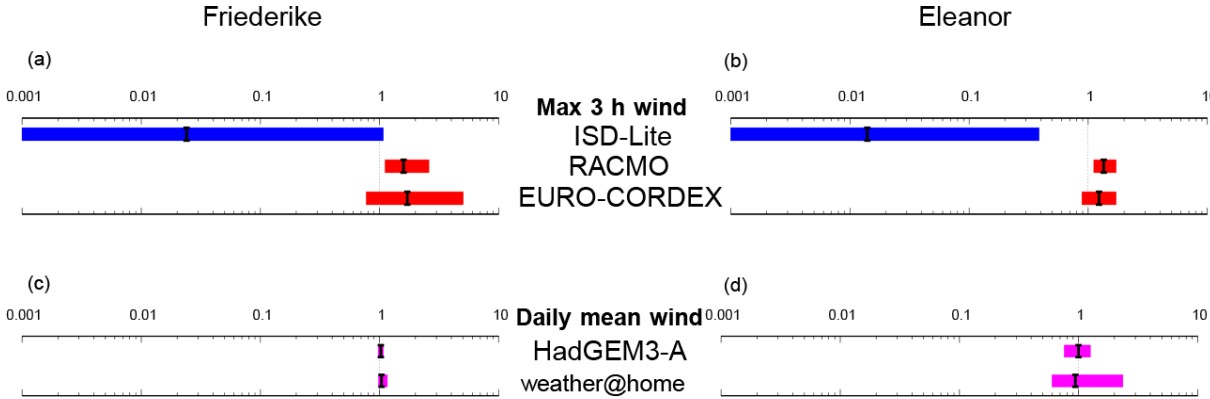

**Figure 10.** Synthesis of the probability ratios (PRs) compared for storms Friederike and Eleanor. Panels **(a)** and **(b)** compare PRs for the daily maximum of three-hourly instantaneous wind speeds (blue for observations and red for models). The PRs have been converted to apply to the common interval 1975–2055 assuming the logarithm scales with the $CO_2$ concentration. Panels **(c)** and **(d)** compare PRs for the two models with daily mean wind speeds. The PRs have been converted to apply to the common interval 1860–2055 to show the change that occurs between pre-industrial and 1.5 °C conditions.

der Wiel, 2017), but this would necessitate using a model result for the future, which would be model-dependent. We therefore use the observed/RCP4.5 $CO_2$ concentration for $f(\text{yr})$. This measure correlates very well with the global mean temperature on the observed record ($r = 0.94$) and permits extrapolation of the probability ratios into the future.

In contrast to the observations, global and regional climate models do not simulate a decrease over the past decades. Instead, simulations of the daily maximum of three-hourly instantaneous wind, of the same spatial and temporal characteristics of these storms and, hence, the observational analysis, indicate increases in probability between 1975 and 2055, corresponding to increases in wind speed for this return period (Fig. 10). These are not all significantly different from 1, but model consensus and future trends support the presence of such a positive tendency. The change is small though: a probability ratio of 1.5 for Friederike with an uncertainty range of 1 to 2, corresponding with an increase in intensity of the wind of only about 5 % (0 % to 10 %). For Eleanor the numbers are even smaller: an increase in PR of about 1.25 (1.0 to 1.6) or an increase in intensity of 2 % (0 % to 5 %).

The changes in daily mean wind are smaller still and indistinguishable from no change. However, as these do not correspond directly to the impact of these storms, we do not take them into account in the synthesis.

The climate model simulations do not always include changes in aerosols and either have no roughness changes (e.g. regional models) or capture these only partially (e.g. HadGEM3-A). This explains at least partially the conflict with the observed trends, as the potential wind results for the Netherlands showed that roughness plays a large role in the observed decrease in storminess. By contrast, these model ensembles mainly reflect changes due to greenhouse gases.

We conclude that storms like Friederike and Eleanor have not become significantly more or less frequent due to climate change, but our model results indicate that global warming due to greenhouse gases could make storms like them somewhat more frequent in the future, with a frequency increase up to at most a factor of 2, or equivalently a few percent higher wind speeds. However, this may seem contradictory with the observations showing a clear and significant decline in high wind speeds in accordance with earlier studies. This is equivalent to declining probabilities of these kinds of storms, but our analysis and previous studies find explanations for these changes in factors other than greenhouse gases. The increase in surface roughness due to forest growth and urbanization potentially explains a major part of this decrease (Vautard et al., 2010; Wever et al., 2012), and does not exclude other factors, such as decadal variability and aerosol effects. Until a quantitative attribution of past observed decreases is established – and with that an understanding of the interplay between greenhouse gas forcing and those other factors, and scenarios for them – the confidence on future evolutions of wind storms will remain low, based on simulations reflecting mainly the effects of greenhouse gas increases. This comes in addition to the poor understanding of how atmospheric circulation variability changes (Shepherd, 2014).

We finally note that changes in extreme probabilities, in general, can be due to both dynamical changes (changes in atmospheric circulation types) and thermodynamical changes (changes in temperature, its gradient, humidity, etc.) due to anthropogenic factors. Our analyses do not attempt to discern the exact origin and processes of changes as it would involve a completely different set of analyses such as done in Vautard et al. (2016) or Yiou et al. (2017). While for temperature and precipitation extremes such a distinction between dynamical and thermodynamical processes is relatively straightforward, we do not think such a separation

would be as simple to interpret here, precisely because the mechanisms are coupled for winds.

**Data availability.** The time series of the annual maximum of area averaged daily maximum wind speed for both storms in the ISD-Lite observations, bias-corrected EURO-CORDEX ensemble, and the RACMO ensemble are available for download and analysis at https://climexp.knmi.nl/selectfield_att.cgi TS5 .

**Author contributions.** RV, GJvO, FELO, and CT set up the research question. RV did most of the data extraction and analysis, GJvO performed the observational analysis and checked the model analysis, FEL did the weather@home analysis. HdV provided Fig. 1 and EvM the RACMO data. AS provided the Dutch wind data. J-MS provided meteorological information and RS wrote the impacts section. All authors contributed text to the article.

**Competing interests.** The authors declare that they have no conflict of interest.

**Acknowledgements.** This work was supported by the EU-PHEME project, which is part of ERA4CS, an ERA-NET initiated by JPI Climate and co-funded by the European Union (grant no. 690462). It was also supported by the French Ministry for an Ecological and Solidary Transition through national convention on climate services and ERC grant no. 338965-A2C2. We would like to thank all volunteers who have donated their computing time to weather@home. The work was initially published by the same authors on the World Weather Attribution web site: https://www.worldweatherattribution.org/ (last access: 16 March 2018) as a "rapid attribution study", and this article is taking most of the material from this analysis and refined it.

**Review statement.** This paper was edited by Christian Franzke and reviewed by three anonymous referees.

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

## Remarks from the language copy-editor

## Remarks from the typesetter

**TS3** I could not find the values mentioned in the following two comments: "Table 2, first line, the return period of ISD-Lite should not be blank but "13.5 [8-400]" year and the PR for the current climate "4 [1-100]" (as mentioned in the text of section 5.1)." and "Table 3: first line, the return period of ISD-Lite should not be blank but "20 [6-200]" and the PR should be "8 [1.5-100]" as mentioned in the text of section 6.1.". However, please note that changing values at this stage needs to be approved by the editor before inserting them.