# Peer review of "Human influence on European winter wind storms such as those of January 2018"

_Earth System Dynamics, 2018_

## Short Comment (SC1) · 10 Oct 2018

The correct reference for the HadGEM3-A modelling system and data is

https://doi.org/10.1016/j.wace.2018.03.003

Ciavarella, Andrew, Nikos Christidis, Martin Andrews, Margriet Groenendijk, John Rostron, Mark Elkington, Claire Burke, Fraser C. Lott, and Peter A. Stott. "Upgrade of the HadGEM3-A based attribution system to high resolution and a new validation framework for probabilistic event attribution." Weather and Climate Extremes (2018).

---

## Referee Comment (RC1) · Paolo De Luca (Referee) · 20 Oct 2018

General comments:

The manuscript addresses a very exciting field of research and it does this with two case studies. The motivations are valid and the attribution analysis on storminess over Europe is clear, well defined, robust and significant for many socio-economic aspects. I suggest minor revisions.

Specific comments:

You will find all these comments in the PDF file attached to my review. They mainly concern a few not clear concepts, Figures' design and statistical tests that have not been mentioned in the text.

[Figure]

In particular:

i) since there are many figures in the text, I suggest to group some of them into one (i.e. 5 to 6 and 10 to 12) so that the number will be reduced and the overall flow kept tighter;

ii) it is also not clear to me which statistical test you performed for checking the difference between the climate periods (e.g. pag. 8 line 22);

iii) there is no reference in the text with respect to Table 2 and 3;

iv) I like the discussion about surface roughness. However it would be nice to have some example in paragraph 6 about what caused the increase in roughness you mentioned;

v) please also double-check all references.

Technical corrections:

You will find all the technical corrections in the PDF file attached to my review.

I suggest to go directly through the comments in the PDF using the latest version of Adobe Acrobat Reader.

pdl

Please also note the supplement to this comment:
https://www.earth-syst-dynam-discuss.net/esd-2018-57/esd-2018-57-RC1-supplement.pdf

**Supplement:**

[revised manuscript text omitted]

---

## Referee Comment (RC2) · Anonymous Referee #2 · 5 Nov 2018

General comments:

This manuscript compares observed trends in extreme values of near surface wind speed over specific areas in Europe with trends predicted by climate models. This is an important topic that fits well with the scope of Earth System Dynamics journal. The observed trends are calculated based on measurements from stations located in unevenly spaced locations over Europe. For the statistical analysis of the observations the authors use the data from all stations in two specific latitude-longitude boxes, which roughly include the main areas affected by two specific storms during January 2018. For the analysis of trends in climate models, the authors use the same boxes used for the observational analysis, but the data is from the evenly spaced grid points in the box. The trends in four different ensembles of climate models are compared. The

trends are evaluated using a numerical fit of the data to a generalized extreme value function (GEV). The results presented in this manuscript show a contradiction between the negative trend in extreme values of near surface wind speed seen in observations and the positive trend predicted by climate models. The authors suggest that this contradiction arises from factors that are not taken into account in climate models such as changes in the surface roughness and aerosol concentration. Though the results are interesting and the mathematical analysis is appropriate for the purpose of the research question, I think some choices in the analysis methods are not well justified. Specifically, the choice of the boxes is not well justified, nor is any sensitivity test presented for this choice. In addition, I found the text and some of the figures to be unclear in several places of the manuscript and some of the data are not properly explained. I therefore recommend on major revisions, according to the comments below.

Specific comments:

1) The focus on the two storms "Friederike" and "Eleanor" is understandable from the point of view of the motivation. However, I see no added scientific value in using the specific locations and maximum wind speeds of these storms as the criteria for comparing observed trends in wind speed with the predicted model trends. Instead, the comparison could be made over various location in central Europe and referring to general trends in extreme wind speed. This would justify the more general conclusions drawn from the results. 2) In accord with comment 1, I suggest shortening section 2, which describes the storms of January 2018 in detail. Figures 1 and 2, which are discussed in this section, describe the intensity of these storms and the specific meteorological conditions that prevailed during that stormy month. These conditions are not discussed any further and are not related to the analysis in the rest of the paper or to the conclusions. I therefore suggest to remove them (especially figure 2). 3) Figure 3 shows the locations of the stations from which the observed data is derived, as well as the boxes chosen for the analysis and the values of the maximum wind speed during the two storms. I didn't find in the text a justification for the choice of the boxes,

except for a general statement that wind speeds were largest in these boxes for the respective storm. The boxes are also indicated in figure 1, however, they do not exactly cover the regions of strongest wind speeds. My two concerns regarding the use of the station-based data statistics over the boxes are: a) Whether a different choice of the boxes would change the results and the conclusions. b) Whether the station-based data is comparable with the grid-based data of the models. I therefore suggest that the authors add a sensitivity test to justify this choice. 4) Table 1 is not organized in a clear way. there are two titles for each column, and it is not clear which numbers in the cells refer to which title, not are the initials in the titles defined. Also, not all the initials of the model ensembles are defined in the text. 5) It is not explained what the KNMI dataset is based on (perhaps satellite measurements?) and for which of the analysis shown in the manuscript it is used. 6) The four ensembles are presented without an explanation for the choice of ensembles and what each of them contributes in addition to the others in terms of the goals and conclusions. 7) Line 28 page 7: The meaning of "winter maximum of the daily maximum of three-hourly maximum wind" is not clear. 8) The description of figure 4a (page 7) refers to the x-axis as time, but it is actually temperature anomaly. It is not explained how the time series is converted to a function of temperature and for what purpose. 9) Line 2 in page 8: what do the values in the parenthesis mean? what is CI? Also, in many other places in the manuscript there are values in the parenthesis without any explanation of their meaning. 10) Lines 20-21 on page 8: "The observed indicator value... has a present return period of about 13 years... which is longer than for the observations". The phrasing is confusing. If by "present" the authors refer to the simulations of the current climate then this should be mentioned explicitly. Secondly, it is mentioned earlier that the return period in observations is about 20 years, which is contradictory to this statement. 11) Figure 5: the caption doesn't explain what the dashed lines indicate. 12) Line 23, page 8: Define "probability ratio". What are the two probabilities for which the ratio is calculated? 13) Last sentence in page 8: the comment about the increase of variability is not clear. 14) The analysis of the HadGEM3A ensemble: It is explained that since this model only

produces daily mean values it is difficult to detect storms such as "Friederike" with this model's data. If so, it would make more sense to either: a) examine the general trends in extreme wind speed, without referring to this specific storm. b) removing this data from the manuscript. The same goes for the weather@home data. 15) Line 12, page 9: It says that in the EURO-CORDEX ensemble the PR is generally not significantly different from one. It is not clear what this statement is based on. The PR values in figure 7 are similar to those in the figure 5 for the RACMO ensemble, for which the authors did identify values significantly different from one. 16) Discussion of figure 14: The choice to scale the probability with $CO_2$ concentration instead of temperature is not well justified or discussed in the text. 17) Section 6: There is not enough discussion of the problems in comparing the observations and model data. Specifically the consequence of having a very different concentration of data points in the two sets is not discussed and the fact that the observations end at 2018, while the climate models continue to much longer periods (or predictions) with a much stronger climate change signal is not discussed.

Technical corrections:

1) Line 29 on page 5: I assume that the second future period was supposed to be [2041-2070]. 2) Line 5 on page 6: the HistoricalNat ensemble is not defined (maybe these are the same simulations defined as natural/counterfactual?). 3) "Global mean temperature" is used in line 29 in page 6 and in figures 4 and 9, instead of "global mean temperature anomaly". 4) The font size of the figure labels and legend is too small. 5) Caption of figure 12: change to "same as figure 9".

---

## Referee Comment (RC3) · Anonymous Referee #3 · 6 Nov 2018

Review result of "Human influence on European winter wind storms such as those of January 2018" by Vautard et al.

Overall recommendation: Major revision

Although this manuscript is interesting, I recommend a major revision because there are many some clarity and logic issues. Below I have three major issues about this manuscript. This needs a further clarification by the authors even though solving the issues is difficult.

Major comments:

(1) I don't know whether these models and which one can better simulate strong wind events or storms over Europe. If ok, these models can used to examine how Human activity affects the European winter wind storms. If no, these results are not real.

(2) It is well known that the storm changes are modulated by the basic air temperature gradient and atmospheric circulation patterns. The authors should tell the reader how changes in storms are linked to basic air temperature gradient and atmospheric circulation pattern changes. Thus, I think that some changes of European wind storms are induced by atmospheric internal variability such as NAO and North Atlantic jet variability. I don't see any information about the internal variability of European winter wind storms from this manuscript.

(3) I think that it is important to differentiate the human influence and internal variability of wind storms over Europe. In fact, the human activity does not only affect the background atmospheric temperature gradient, but also on the atmospheric circulation patterns. Because the human influence and internal variability are coupled together, it is difficult to say the human activity can have a significant influence on European wind storms even though the authors presented many model results.

[Figure]

Minor comments:

(1) Page 1, line 20: "covering The Netherlands" should be "covering the Netherlands".

(2) Page 2, line 5: "By contrast, A more zonal flow" should be "By contrast, a more zonal flow".

(3) The circulation patterns in Fig.1 are different from those in Fig. 2. In fact, the circulation patterns in Fig. 1 are wave trains that are comprised of European blocking and NAO+ patterns (Luo et al. 2015, JC, 6398-6418). Thus, it may be difficult to use the circulation patterns in Fig. 2 to account for the winter wind storms over Europe.

---

## Author Comment (AC1) · 1 Feb 2019

The reference has been changed

———————————————————

---

## Author Comment (AC2) · 1 Feb 2019

We thank the reviewer for comments and suggestions. The rebuttal is attached, with response to all reviewers

Please also note the supplement to this comment:
https://www.earth-syst-dynam-discuss.net/esd-2018-57/esd-2018-57-AC2-supplement.pdf

---

## Author Comment (AC3) · 1 Feb 2019

**Rebuttal**

Reviewer #1

General comments:

The manuscript addresses a very exciting field of research and it does this with two case studies. The motivations are valid and the attribution analysis on storminess over Europe is clear, well defined, robust and significant for many socio-economic aspects. I suggest minor revisions.

We thank the reviewer for the thorough review and comments, including the supplement which includes a useful number of editorial comments.

Specific comments:

You will find all these comments in the PDF file attached to my review. They mainly concern a few not clear concepts, Figures' design and statistical tests that have not been mentioned in the text.

In particular:

i) since there are many figures in the text, I suggest to group some of them into one (i.e. 5 to 6 and 10 to 12) so that the number will be reduced and the overall flow kept tighter;

The figures have been grouped according to the reviewer suggestion.

ii) it is also not clear to me which statistical test you performed for checking the difference between the climate periods (e.g. pag. 8 line 22);

The bootstrap procedure for confidence interval estimation is now described in the end of Section 4.

iii) there is no reference in the text with respect to Table 2 and 3;

We now reference these tables

iv) I like the discussion about surface roughness. However it would be nice to have some example in paragraph 6 about what caused the increase in roughness you mentioned;

The urbanization and the forest growth as assumptions have been cited in the text of the synthesis section (now Section 8), with the references having some discussions.

v) please also double-check all references.

References have now been checked and updated

Technical corrections:

You will find all the technical corrections in the PDF file attached to my review. I suggest to go directly through the comments in the PDF using the latest version of Adobe Acrobat Reader.

Almost all suggestions have been taken into account. One important was the statistical testing, which is now described in Section 4.

Reviewer #2

General comments:

This manuscript compares observed trends in extreme values of near surface wind speed over specific areas in Europe with trends predicted by climate models. This is an important topic that fits well with the scope of Earth System Dynamics journal. The observed trends are calculated based on measurements from stations located in unevenly spaced locations over Europe. For the statistical analysis of the observations the authors use the data from all stations in two specific latitude-longitude boxes, which roughly include the main areas affected by two specific storms during January 2018. For the analysis of trends in climate models, the authors use the same boxes used for the observational analysis, but the data is from the evenly spaced grid points in the box. The trends in four different ensembles of climate models are compared. The trends are evaluated using a numerical fit of the data to a generalized extreme value function (GEV). The results presented in this manuscript show a contradiction between the negative trend in extreme values of near surface wind speed seen in observations and the positive trend predicted by climate models. The authors suggest that this contradiction arises from factors that are not taken into account in climate models such as changes in the surface roughness and aerosol concentration. Though the results are interesting and the mathematical analysis is appropriate for the purpose of the research question, I think some choices in the analysis methods are not well justified. Specifically, the choice of the boxes is not well justified, nor is any sensitivity test presented for this choice. In addition, I found the text and some of the figures to be unclear in several places of the manuscript and some of the data are not properly explained. I therefore recommend on major revisions, according to the comments below.

We thank the reviewer for the detailed review and remarks. We hope the revised version of the article has addressed all the reviewer's comments.

Specific comments:

1) The focus on the two storms "Friederike" and "Eleanor" is understandable from the point of view of the motivation. However, I see no added scientific value in using the specific locations and maximum wind speeds of these storms as the criteria for comparing observed trends in wind speed with the predicted model trends. Instead, the comparison could be made over various location in central Europe and referring to general trends in extreme wind speed. This would justify the more general conclusions drawn from the results.

The analysis of specific events in a relatively narrow class around observed events is exactly what "event attribution" attempts to achieve, as compared to more general studies of a larger class of events over a large continental region (see eg. Stott et al., 2016). We agree that the latter type of studies is also necessary but what is intended in "event attribution" is precisely to focus on the event that just occurred, in order to put what just occurred in the context of climate change. This is motivated by several reasons, among which the general question of the public of the link with climate change when a specific event occurs, or potential links with legal issues.

We elaborated on this by rephrasing and adding sentences in the introduction.

In addition to this, and to further address the reviewer's question, we conducted a sensitivity analysis on the domain definition, which is now reported in a Section 7. This sensitivity analysis does not attempt to generalize our results to all European winter storm, but rather to strengthen the robustness of our conclusions for the storms Friederieke and Eleanor. From

this analysis, carried out only with the RACMO ensemble, we show that our conclusions also hold when changing the domain definition.

2) In accord with comment 1, I suggest shortening section 2, which describes the storms of January 2018 in detail. Figures 1 and 2, which are discussed in this section, describe the intensity of these storms and the specific meteorological conditions that prevailed during that stormy month. These conditions are not discussed any further and are not related to the analysis in the rest of the paper or to the conclusions. I therefore suggest to remove them (especially figure 2).

In the perspective of describing these specific events and their impacts and to put them in the context of climate change, rather than a broader class of events, we believe it is important to keep the description of the events and their consequences. This is a purely descriptive section, which places the context rather than be used in the statistical analysis.

3) Figure 3 shows the locations of the stations from which the observed data is derived, as well as the boxes chosen for the analysis and the values of the maximum wind speed during the two storms. I didn't find in the text a justification for the choice of the boxes, except for a general statement that wind speeds were largest in these boxes for the respective storm. The boxes are also indicated in figure 1, however, they do not exactly cover the regions of strongest wind speeds. My two concerns regarding the use of the station-based data statistics over the boxes are: a) Whether a different choice of the boxes would change the results and the conclusions. b) Whether the station-based data is comparable with the grid-based data of the models. I therefore suggest that the authors add a sensitivity test to justify this choice.

We followed the recommendation of the reviewer and added a sensitivity analysis which is now reported in the new Section 7. In this analysis, carried out for one model ensemble (RACMO) for the sake of conciseness, we use two new domain definitions for each of the two storms, with the following characteristics: one extended domain and one reduced domain, centered on areas with high winds. We show in this sensitivity analysis that in each case we find similar results, which shows the robustness of our main conclusions.

4) Table 1 is not organized in a clear way. there are two titles for each column, and it is not clear which numbers in the cells refer to which title, not are the initials in the titles defined. Also, not all the initials of the model ensembles are defined in the text.

We have added several details in the legend of Table 1 in order to make the table description fully understandable.

5) It is not explained what the KNMI dataset is based on (perhaps satellite measurements?) and for which of the analysis shown in the manuscript it is used.

The KNMI data are in-situ observations at weather stations in the country. We have added this information to the description. They are used in section 5.1, last paragraph.

6) The four ensembles are presented without an explanation for the choice of ensembles and what each of them contributes in addition to the others in terms of the goals and conclusions.

Except for weather@home, model simulations were not carried out specifically for this study. The ensembles were selected, as usual in attribution studies, according to climate simulations available to the authors that can be used for attribution studies, taking into account the resolution required to represent the extremes under study. HadGEM3-A and weather@home have the advantage of having counterfactual simulations, using SSTs where the anthropogenic signal is removed, whereas RACMO and EURO-CORDEX transient

simulations were carried out at a higher resolution allowing extremes such as wind storms to be better captured. We added the following text in Section 4:

We used four climate model simulation ensembles. These ensembles are complementary: two of them are made of regional climate simulations downscaling low-resolution global climate models (GCMs) with a high-resolution (12.5 km). One of these latter is using the same model chain with different members for the GCM, while the second one is a multi-model ensemble member. We therefore cover several aspects of the uncertainty. The other two ensembles were available at the time of the study and also used, one of which consists of a very large ensemble. However for these latter only daily mean wind speed was available while for the former daily maximum wind speed was available. Our assessment is therefore based on the first two ensembles, which better represent the January 2018 storms. The other two ensembles are used for consistency checking.

7) Line 28 page 7: The meaning of "winter maximum of the daily maximum of three-hourly maximum wind" is not clear.

We have rephrased this sentence to: "We compute the daily maximum of three-hourly wind speed (taken at 00, 03, 06, 09, 12, 15, 18 and 21 UTC) at each station and averaged it over the ISD-lite stations available in the box. The winter maximum of this quantity is shown…"

8) The description of figure 4a (page 7) refers to the x-axis as time, but it is actually temperature anomaly. It is not explained how the time series is converted to a function of temperature and for what purpose.

We had done the analysis both as a function of time and as a function of the global mean temperature, and accidentally inserted the figure from the other analysis (the results are very similar). This has been corrected.

9) Line 2 in page 8: what do the values in the parenthesis mean? what is CI? Also, in many other places in the manuscript there are values in the parenthesis without any explanation of their meaning.

We have explicited the mentions to confidence intervals with: "The decrease in intensity of about 12% with a 95% confidence interval of 0–30%, which we will denote thereafter (95% CI: 0-30%)…"

10) Lines 20-21 on page 8: "The observed indicator value: : : has a present return period of about 13 years: : : which is longer than for the observations". The phrasing is confusing. If by "present" the authors refer to the simulations of the current climate then this should be mentioned explicitly. Secondly, it is mentioned earlier that the return period in observations is about 20 years, which is contradictory to this statement.

We thank the reviewer for pointing out the confusion. The "20 year" was an order of magnitude, but the value obtained from the numbers directly is 13.5 years as shown in Table 1. We removed the confusion by rephrasing and letting numbers to be coherent with Table 1.

11) Figure 5: the caption doesn't explain what the dashed lines indicate.

This is now explained.

12) Line 23, page 8: Define "probability ratio". What are the two probabilities for which the ratio is calculated?
We now introduce thresholds, probabilities and PRs in the beginning of Section 3:

Classical event attribution relies on defining an event as an exceedance of a threshold in the tail of the distribution of an event indicator. Once the event is defined the probability of exceeding the threshold is calculated for the current climate and for an hypothetical climate where anthropogenic influence is not present or largely reduced. Once this is done, the ratio of the probabilities (probability ratio denoted herefater as "PR") is estimated. We define here the indicators associated to the two studied storms.

13) Last sentence in page 8: the comment about the increase of variability is not clear.

We removed the confusing end of the sentence.

14) The analysis of the HadGEM3A ensemble: It is explained that since this model only produces daily mean values it is difficult to detect storms such as "Friederike" with this model's data. If so, it would make more sense to either: a) examine the general trends in extreme wind speed, without referring to this specific storm. b) removing this data from the manuscript. The same goes for the weather@home data.

We agree that the information brought by these two model ensembles is weaker given that they do not represent instantaneous winds. However we feel that they complement the analysis of the other models by showing that their results for the daily time scale are in agreement with the RACMO and EURO-CORDEX results. We therefore prefer to keep these model data in the article. In order to place daily results at a different level than the results on daily wind speed maxima, we have reordered the presentation of results using first RACMO, then EURO-CORDEX, then HadGEM and weather@home.

15) Line 12, page 9: It says that in the EURO-CORDEX ensemble the PR is generally not significantly different from one. It is not clear what this statement is based on. The PR values in figure 7 are similar to those in the figure 5 for the RACMO ensemble, for which the authors did identify values significantly different from one.

These statements are based on the values of the low confidence interval boundary in Figures 5 and 7. In both cases the significance is marginal: the C.I. is slightly above 1 for RACMO and slightly below 1 for EURO-CORDEX. We agree with the reviewer and we have rephrased our statements to make them more consistent.

16) Discussion of figure 14: The choice to scale the probability with CO2 concentration instead of temperature is not well justified or discussed in the text.

We have expanded the discussion of the choice of our indicator of global warming in the paragraph underneath the corresponding equation.

17) Section 6: There is not enough discussion of the problems in comparing the observations and model data. Specifically the consequence of having a very different concentration of data points in the two sets is not discussed and the fact that the observations end at 2018, while the climate models continue to much longer periods (or predictions) with a much stronger climate change signal is not discussed.

This discussion is now developed by explicitly describing the methods used to process both models and observations, and we explain how results from observations and models can be compared, due to the time-dependent parametric fit done using observations.

Technical corrections:

1) Line 29 on page 5: I assume that the second future period was supposed to be [2041-2070].

Corrected

2) Line 5 on page 6: the HistoricalNat ensemble is not defined (maybe these are the same simulations defined as natural/counterfactual?).

Changed to "counterfactual"

3) "Global mean temperature" is used in line 29 in page 6 and in figures 4 and 9, instead of "global mean temperature anomaly".

This is changed in the text but for figures the time is now used.

4) The font size of the figure labels and legend is too small.

5) Caption of figure 12: change to "same as figure 9".

Note that figures have been grouped, and legends have changed accordingly. Some extra labelling has been added and we hope the visibility will be improved by the production as pdf.

Reviewer #3

Overall recommendation: Major revision

Although this manuscript is interesting, I recommend a major revision because there are many some clarity and logic issues .Below I have three major issues about this manuscript.This needs a further clarification by the authors even though solving the issues is difficult.

We thank the reviewer for the review and we hope we have addressed all the reviewer's concerns.

Major comments:

(1) I don't know whether these models and which one can better simulate strong wind events or storms over Europe. If ok, these models can used to examine how Human activity affects the European winter wind storms. If no, these results are not real.

The issue of how models do simulate winds and in particular strong winds in the area of the storms is addressed in detail by comparing the storm indicators with both reanalysis data and station data in Section 4. The comparison between models and observations of the daily wind maximum is made in Table 1. The WX99 index indicates underestimation of the models of about 15-20%. For HadGEM3-A and weather@home ensembles it is difficult to conclude of a capacity to simulate such storms as only daily mean winds were available. However strong daily mean winds are compared. Due to this issue, our conclusions are based essentially on RACMO and EURO-CORDEX.

(2) It is well known that the storm changes are modulated by the basic air temperature gradient and atmospheric circulation patterns. The authors should tell the reader how changes in storms are linked to basic air temperature gradient and atmospheric circulation pattern changes. Thus, I think that some changes of European wind storms are induced by atmospheric internal variability such as NAO and North Atlantic jet variability. I don't see any information about the internal variability of European winter wind storms from this manuscript.

We believe this question, which would need a general investigation of the changes in dynamical processes in the mid-latitude, goes well beyond the scope of this article. This question would address the general issue of overall changes in all storms in Europe in a broader context, which has also been addressed in a few cited studies. The specificity of this article is to assess potential links between the specific storms studied and climate change. However we investigated how aspects of the variability are changing in Figure 2 by looking at frequencies of the main European weather regimes in winter (see Figure 2). From this figure we could not find significant changes in the main weather regimes. However we know that the main atmospheric circulation is changing with a poleward displacement of the jet streams. The work of Li et al. (2018) is now cited in the introduction.

(3) I think that it is important to differentiate the human influence and internal variability of wind storms over Europe. In fact, the human activity does not only affect the background atmospheric temperature gradient, but also on the atmospheric circulation patterns. Because the human influence and internal variability are coupled together, it is difficult to say the human

activity can have a significant influence on European wind storms even though the authors presented many model results.

The attribution techniques are precisely devoted to discern changes (altogether) due to human influence from changes due to natural variability as found in a natural world, by examining the differences between factual, observed or simulated probabilities in the current climate and counterfactual simulations, or simulations differing by the greenhouse gases forcing. Such is the case of all simulations here. We agree that internal variability can also change due to anthropogenic factors, and also result from thermo-dynamical changes. Our analyses do not attempt to discern the exact origin and processes of changes as it would involve a completely different set of analyses such as done in Vautard et al. (2016). While for temperature and precipitation such a distinction between dynamical and thermodynamical processes is relatively straightforward, we do not think here such a separation would be as simple to interpret, because precisely the mechanisms are completely coupled for winds, as noted by the reviewer. We feel such an analysis is well beyond the scope here and would deserve an entirely new research project. A short paragraph has been added in the conclusion for this.

Minor comments:
(1) Page 1, line20: "covering The Netherlands" should be "covering the Netherlands".

Done

(2) Page 2, line 5:"By contrast, A more zonal flow" should be "By contrast, a more zonal flow".

Done

(3) The circulation patterns in Fig.1 are different from those in Fig. 2. In fact, the circulation patterns in Fig. 1 are wave trains that are comprised of European blocking and NAO+ patterns (Luo et al. 2015, JC, 6398-6418). Thus, it may be difficult to use the circulation patterns in Fig. 2 to account for the winter wind storms over Europe.

Since Figure 1 does not represent circulation patterns but wind speed maxima, we are assuming that the reviewer is comparing Fig. 2 a-b with Figure 2 (medium row). The anomaly of SLP in figure 2b, which is a monthly mean, indeed is not related with any of the weather regimes below. However, this is not surprising because weather regimes are characteristic of daily weather patterns in general and not monthly. Fig. 2b is only displayed to describe the mean circulation that was present along the full month of Jan 2018. We do not think these can be compared. The evolution of frequency of weather patterns in Figure 2 is only here to give a simple overview of the (absence of) changes in gross circulation types.